# Polyherbal and Multimodal Treatments: Kaempferol- and Quercetin-Rich Herbs Alleviate Symptoms of Alzheimer’s Disease

**DOI:** 10.3390/biology12111453

**Published:** 2023-11-20

**Authors:** Claire Alexander, Ali Parsaee, Maryam Vasefi

**Affiliations:** 1Department of Biology, Lamar University, Beaumont, TX 77705, USA; 2Biological Science, University of Calgary, Calgary, AB T2N 1N4, Canada

**Keywords:** Alzheimer’s disease (AD), kaempferol, quercetin, flavonoids, traditional Chinese medicine, dementia

## Abstract

**Simple Summary:**

Despite the well-documented pathophysiology of Alzheimer’s Disease (AD), treatment options are limited in diversity and efficacy. Thus, the development of new treatments requires an extensive understanding of molecular pathways altered by drugs in development. In this review, we survey the literature regarding common herbal phytochemicals, kaempferol and quercetin, with a specific focus on their multiple mechanisms that alleviate the pathological underpinnings of AD. Here, we utilize the well-documented mechanisms of quercetin to propose a novel multimodal mechanism of kaempferol, and we discuss common herbal sources and the limitations of these potential treatments.

**Abstract:**

Alzheimer’s Disease (AD) is a progressive neurodegenerative disorder impairing cognition and memory in the elderly. This disorder has a complex etiology, including senile plaque and neurofibrillary tangle formation, neuroinflammation, oxidative stress, and damaged neuroplasticity. Current treatment options are limited, so alternative treatments such as herbal medicine could suppress symptoms while slowing cognitive decline. We followed PRISMA guidelines to identify potential herbal treatments, their associated medicinal phytochemicals, and the potential mechanisms of these treatments. Common herbs, including *Ginkgo biloba*, *Camellia sinensis*, *Glycyrrhiza uralensis*, *Cyperus rotundus*, and *Buplerum falcatum*, produced promising pre-clinical results. These herbs are rich in kaempferol and quercetin, flavonoids with a polyphenolic structure that facilitate multiple mechanisms of action. These mechanisms include the inhibition of Aβ plaque formation, a reduction in tau hyperphosphorylation, the suppression of oxidative stress, and the modulation of BDNF and PI3K/AKT pathways. Using pre-clinical findings from quercetin research and the comparatively limited data on kaempferol, we proposed that kaempferol ameliorates the neuroinflammatory state, maintains proper cellular function, and restores pro-neuroplastic signaling. In this review, we discuss the anti-AD mechanisms of quercetin and kaempferol and their limitations, and we suggest a potential alternative treatment for AD. Our findings lead us to conclude that a polyherbal kaempferol- and quercetin-rich cocktail could treat AD-related brain damage.

## 1. Introduction

Alzheimer’s disease (AD) is a debilitating neurodegenerative disorder characterized by cognitive decline and memory impairment. AD could affect 152 million individuals by 2050 [1]. The progression of AD is influenced by multiple factors, including the accumulation of beta-amyloid plaques (Aβ) and the formation of neurofibrillary tangles (NFTs). The aggregation of Aβ plaques exacerbates the disease by impairing neuronal function and triggering neuroinflammation [2,3,4,5]. Oxidative stress and the presence of neurofibrillary tangles (NFTs) also contribute to the aggregation of Aβ into senile plaques [6,7,8,9,10,11,12,13,14]. NFTs consist of hyperphosphorylated tau proteins that disrupt neuronal transport systems [15,16,17,18,19,20,21]. Neuroinflammation, in turn, exacerbates damage to neuronal integrity [22]. Symptoms of AD include memory loss, impaired learning, emotional changes, cognitive and speech deficits, shortened attention span, and impaired management of daily tasks [23,24,25].

Currently, the treatments available for AD are expensive and have minimal efficacy. Acetylcholinesterase inhibitors (AChEIs), including donepezil, and N-methyl-D-aspartate (NMDA) receptor antagonists, including memantine, are commonly prescribed for AD [26]. AChEIs inhibit the enzymatic degradation of ACh by inhibiting cholinesterase activity [27], while NMDA receptor antagonists limit calcium influx to prevent glutamate-induced cytotoxic cell death [28]. However, these drugs simply suppress symptoms and fail to halt disease progression [26], and only half of the population positively responds to these current treatments [29,30]. Herbal medicine boasts a well-documented history of safe and effective incorporation into traditional Asian diets [31,32]. Preclinical studies have demonstrated that these herbs can enhance cognitive and memory functions [33,34]. These herbs serve as dependable sources of phytochemicals, such as kaempferol and quercetin, that have limited side effects and could combat Alzheimer’s disease [35,36,37]. Specifically, these flavonoids have anti-inflammatory, neuroprotective, and anti-degenerative effects [33,38,39,40,41,42,43,44,45,46].

The objective of this review is to elucidate the anti-AD mechanisms of kaempferol and quercetin. Here, we present a multimodal mechanism of action for kaempferol and quercetin in the treatment of Alzheimer’s disease (AD). First, both flavonoids exert antioxidant effects, which stabilize cellular function and reduce neuroinflammation. Importantly, they also modulate PI3K/AKT signaling to limit Aβ and tau accumulation in toxic aggregates and enhance neuroplasticity by restoring BDNF signaling. These mechanisms ultimately improve memory and cognitive performance in AD patients. To our knowledge, this review represents the first comprehensive exploration of the literature that collectively shows kaempferol’s potential to counteract both tau and Aβ via modulation of the PI3K/AKT/GSK-3β pathway. Additionally, we propose that a polyherbal cocktail, incorporating sources rich in quercetin and kaempferol, could serve as an effective adjunctive or alternative treatment for AD. Finally, we explore the limitations of quercetin and kaempferol and discuss potential strategies for overcoming these challenges.

## 2. Materials and Methods

We collected data following the PRISMA guidelines for systematic review articles. The articles were sourced from PubMed, ScienceDirect, and Google Scholar, and data collection was conducted up until November 2023. We compiled a relevant list of articles to identify phytochemicals that have been studied for the treatment of Alzheimer’s disease (AD) and their potential to induce therapeutic brain changes related to AD. Our search strategy initially yielded a total of 13,691 papers (13,688 from databases and an additional 3 from other sources). Of these, 2463 studies were screened based on their titles and abstracts, resulting in 378 articles that met the inclusion criteria (Figure 1). We included studies and reviews that explored the anti-AD mechanisms of phytochemicals and those that provided insights into the features of AD. The language was limited to English. Inclusion criteria required that articles discuss topics such as “Alzheimer’s disease”, “herbs”, “kaempferol”, “quercetin”, “inflammation”, “neuroprotection”, “tau”, and “Aβ”. The selected articles encompassed reviews, original research articles, and published clinical trials. Data extraction was carried out independently by a team of three investigators, considering factors such as the year of publication, article types, and the topic of herbs in relation to AD.

## 3. Hallmarks of Alzheimer’s Disease

Several features of AD, including Aβ plaque accumulation [47,48], tau hyperphosphorylation and neuroinflammation [49], and oxidative stress [50,51,52,53], have been identified as targets for drug development. Moreover, these deficits have been observed in studies with human patients [6,54,55,56,57,58,59,60,61,62,63]. This section will briefly explore the pathophysiology of AD, with a focus on the proposed molecular origins and outcomes of their aberrant activities.

While the origins are still debated, the literature greatly supports the roles of oxidative stress and neuroinflammation as critical drivers of neurodegeneration. Antioxidant deficits facilitate ROS production, driving oxidative stress via lipid peroxidation [57]. Consequently, mitochondrial energy production is impaired and pro-apoptotic signaling follows [57]. Glutamate-induced excitotoxicity could also facilitate oxidative stress [64,65,66,67]. Disrupted ROS clearance establishes the neuroinflammatory microglial and astrocytic hyperactivity [38,68,69,70,71,72] and favors neuronal signaling pathways that impair Aβ clearance [48,73,74]. Finally, proper mitochondrial function is required for Aβ clearance and can, in turn, maintain appropriate tau activity states [75].

Although normal Aβ levels can maintain regular neuronal function [76], failed Aβ clearance from the brain can expedite neurodegeneration by facilitating plaque accumulation and impairing neuronal communication [22,47,48,75,77,78,79]. Moreover, Aβ accumulation further promotes oxidative stress [80,81,82,83]. As Aβ plaques accumulate in the brain due to impaired clearance [84], overzealous astrocytic and microglial responses compound the neuroinflammatory environment by releasing pro-inflammatory factors, promoting neuronal apoptosis [6,49,85,86,87,88,89]. These findings were supported in postmortem tissue [6,54,55,56]. Finally, Aβ signaling significantly impairs LTP [90], facilitating neurodegeneration via low synaptic activity.

AD is one of the most common tauopathies [91]. Aβ plaque accumulation drives tau hyperphosphorylation [47,58,92,93,94,95,96,97,98,99,100], possibly by excess GSK-3β signaling [101]. Likewise, tau hyperphosphorylation also compounds Aβ toxicity [102,103], which has been supported by PET imaging in humans with memory impairment and cognitive decline [60]. These studies demonstrate that Aβ toxicity is necessary for tau hyperphosphorylation [59,60]. Specifically, accumulating Aβ binds to NMDAR, generating excess calcium levels to activate calpain-mediated microtubule-associated protein cleavage [65,104,105]. These events impair mitochondrial function, invoking pro-apoptotic signaling [65,106]. Tau hyperphosphorylation dismantles axonal microtubules to degenerate the axon [15,107,108,109,110], impairing synaptic plasticity [102,103,111,112]. Hyperphosphorylated tau spreads throughout the hippocampus in AD models [113], and uptake may be mediated by clathrin-induced endocytosis [114]. Risk factors such as sleep apnea may potentiate the spread of tau in this manner [115]. Ultimately, these events result in neuronal death and compromise neuroplasticity, thereby driving neuroinflammation and impairing cognitive function.

## 4. Anti-AD Mechanisms of Quercetin and Kaempferol

Given the limited therapeutics available to AD patients, it is essential to explore alternative treatments, such as plant-derived phytochemicals. Flavonoids, including kaempferol and quercetin, belong to the class of polyphenols commonly found in various herbs. Notably, kaempferol and quercetin possess lipophilic properties [50], which facilitate their easy entry into cells. These phytochemicals are abundant, with an average daily consumption of approximately 23 mg of flavonoids in a typical diet [116,117]. Kaempferol and quercetin produce several beneficial properties, including anti-inflammatory, antioxidant, anti-Aβ, anti-tau, and pro-neuroplastic effects [37,38,39,57,74,118,119,120,121,122,123,124,125,126,127,128]. Moreover, they have demonstrated cognitive and memory-enhancing effects in animal studies [37]. Consequently, this section aims to delve into the commonly studied effects of these phytochemicals.

### 4.1. Quercetin

Quercetin, the most prevalent flavonoid, is found in several traditional medicinal herbs and is commonly found in fruits and vegetables, including berries, onions, and leeks [118,129,130,131,132,133,134,135,136,137,138,139]. Quercetin intake constitutes approximately 60–75% of total flavonols [140,141], and 25 mg of quercetin is found in the average diet [38]. Quercetin is commonly investigated for its potential anti-neurodegenerative efficacy and is considered safe [51,142]. Quercetin is a 15-carbon flavonoid with two benzene rings connected via a 3-carbon shape (Figure 2) [38,130,143].

Quercetin produces anti-inflammatory effects via multiple signaling pathways, including Nrf2, paraoxonase-2 (PON2), JNK, PKC, and NF-kB [51,118,128,144,145,146,147]. Quercetin dose-dependently protected HT22 hippocampal neurons from glutamate-induced apoptosis by limiting ROS production, impairing the calpain-mediated cleavage of cytoskeletal proteins, and preserving mitochondrial membrane potential [65]. Quercetin also inhibits NO release by inhibiting iNOS activity [33,38,148], which could reduce excess glutamate signaling and minimize the risk of glutamate-induced cytotoxicity in hippocampal neurons in a similar fashion to kaempferol and its derivatives [149]. Moreover, quercetin inhibits COX-2 and TLR4 activity to reduce inflammatory responses [6,39,148]. Interestingly, quercetin may have epigenetic mechanisms by inhibiting lysine acetyltransferase (KAT) activity [150,151] and increasing lysine deacetylase (KDAC) activity [152], suggesting that the flavonoid can bidirectionally regulate autophagy [153], neuroinflammation, and apoptosis [154]. Quercetin also inhibits acetylcholinesterase (AChE) [155], which can enhance alertness and cognitive function in AD patients. 

The anti-Aβ effects of quercetin are well studied in AD and related models and have yielded promising therapeutic properties. The hydrophobic groups of quercetin can inhibit the formation of Aβ fibrils [120,121,122,123,156]. Chronic quercetin treatment also slowed Aβ aggregation by potentiating AMPK signaling and inhibiting mitochondrial ROS production, leading to improved memory and object recognition in APPswe/PS1dE9 [80,157]. Quercetin treatment also inhibits the BACE1-mediated cleavage of APP into Aβ by inhibiting NF-kB [74]. Consequently, mitochondrial membrane permeability is restored, and cellular survival is favored over oxidative stress [158]. This anti-neurodegenerative effect could be due to the free radical-quenching structure of the catechol group, reducing neuroinflammation, lipid peroxidation, mitochondrial stress, and DNA damage [38,51]. Elevated SOD, GPx, and Na^+^-K^+^ ATPase activity could also be due to quercetin’s anti-Aβ effects [44,78].

In many studies, quercetin and its derivatives reduced tau hyperphosphorylation [23,58,132,159]. In rodent HT22 hippocampal neurons, chronic quercetin treatment inhibited tau phosphorylation at four sites by reducing p-Cdk5 levels, limiting calpain activity, and dramatically reducing Ca^2+^ influx [58]. In 3xTgAD mice, chronic quercetin inhibited Aβ pathology, reduced NFT levels, and prevented astrocytic and microglial hyperactivity in the amygdala and hippocampus [132,160], showing that the anti-Aβ and anti-tau mechanisms of quercetin depend on its anti-inflammatory effects. Consequently, these mice demonstrated improved learning and memory and decreased anxiety [132], while combined exercise and quercetin treatment robustly improved spatial memory in AD rodents [161]. Studies also found that quercetin enhanced cell viability and morphology by reducing MDA and ROS levels and increasing antioxidant SOD and GSH activity [159,162], limiting NF-κB signaling, restoring mitochondrial membrane potential to baseline, inhibiting tau hyperphosphorylation, and regulating Akt/PI3K/GSK-3β signaling pathway [159,163]. Taken together, these data show that quercetin has a multimodal mechanism of action in treating AD. Of note, the anti-tau and consequent pro-neuroplastic effect of quercetin is further explored in Section 5, but the primary anti-inflammatory, anti-Aβ, ant-tau, and pro-neuroplastic effects of this flavonoid are all dependent on each other. 

**Figure 2 biology-12-01453-f002:**
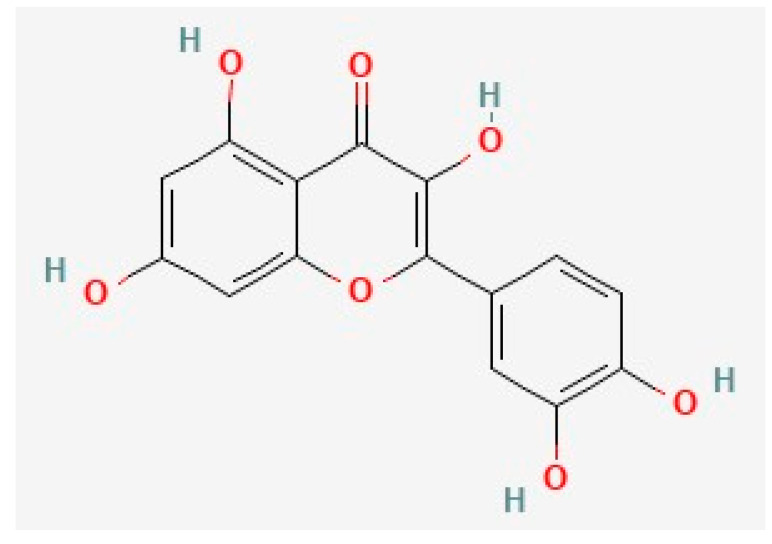
The chemical structure of quercetin, deduced from PubChem [164].

### 4.2. Kaempferol

Kaempferol is a common 15-carbon polyphenol (Figure 3) that shares significant structural similarity with quercetin. It is one of the most common flavonoids and is found in a variety of common foods, including fruits and vegetables [129,130,165,166,167,168,169,170]. Multiple preclinical and clinical studies have supported the anti-AD activity of kaempferol [57,149,171,172,173,174]. Kaempferol has pro-neuroplastic, anti-Aβ, anti-tau, anti-inflammatory, and antioxidant properties [29,44,57,171,175,176,177,178,179]. Notably, kaempferol also inhibits AChE like quercetin [180], but this mechanism is beyond the scope of this review.

Like quercetin, kaempferol and its metabolites reduce inflammation and have potent antioxidant properties [181,182,183,184]. Kaempferol also directly modulates neuroinflammation by impairing microglial TLR4 and NF-kB signaling and inhibiting the release of NO, iNOS, PGE2, IL-1β, TNF-α, and IFN-γ [167,185]. Kaempferol also reversed BBB damage [36,186,187]. Kaempferol can also modulate neuroinflammation by regulating epigenetic factors such as SIRT1, a subtype of KDAC [188,189,190]. Kaempferol also prevents cytotoxic damage to PC12 neurons by upregulating SIRT [191]. Other immune factors modulated by kaempferol include COX-2, lipoxygenases, prostacyclin, and leukotrienes [148,187,192,193,194]. Finally, kaempferol may reduce neuroinflammation via Nrf-2 signaling [185].

Like quercetin, kaempferol and its derivatives reverse Aβ-induced damage [29,120,122,124,125,149,195]. Kaempferol-3-O-rhamnoside (K-3-Rh), a kaempferol derivative, limited total Aβ burden and toxicity by disrupting β-sheet formation and impairing Aβ plaque formation in human SH-SY5Y cells [195,196]. However, kaempferol antagonized fibrilization with lower potency compared to quercetin and morin [120,122]. In rodent neuroblastoma cells, kaempferol 3-O-(6″-acetyl)-β-glucopyranoside (KAG) robustly inhibited Aβ-mediated cytotoxic cell death and ROS generation [149]. KAG reversed Aβ-mediated oxidative stress and increased cell survival by regulating caspase-3, Bax, and Bcl-2 signaling [44,64,149,197,198,199,200]. Kaempferol dose-dependently and sex-dependently limited Aβ-induced mitochondrial toxicity in neurons, improving rodent memory in the Y-maze test [57,134,201]. Of note, studies regarding kaempferol’s direct influence on tau are limited; thus, more research is necessary. However, due to its similar phenolic structure to quercetin [165,166], we hypothesize that kaempferol could also reduce tau hyperphosphorylation.

**Figure 3 biology-12-01453-f003:**
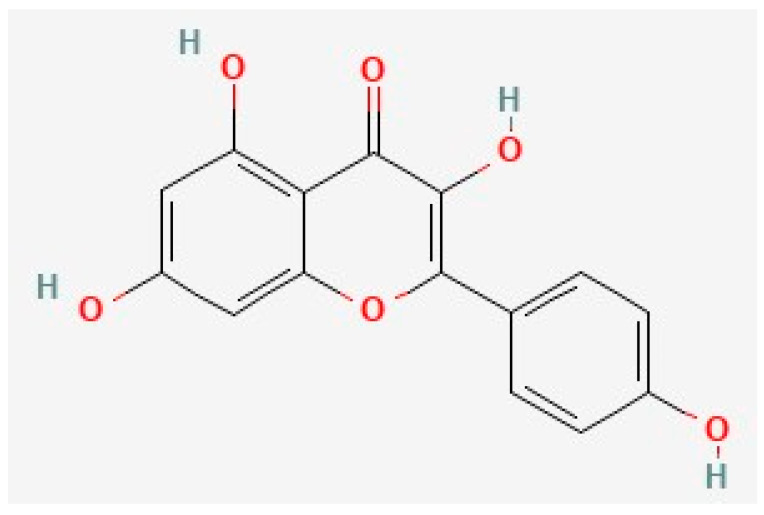
The chemical structure of kaempferol, deduced from PubChem [202].

## 5. Kaempferol, Quercetin, and Neuroplasticity

The aberrant brain changes described in Section 3 can impair memory and cognitive function by creating deficits in neuroplasticity. Thus, future AD treatments should also be designed to directly target signaling pathways that can counteract the etiologies of AD. Specifically, we identified the PI3K/AKT signaling pathway as a critical candidate to counteract neurodegeneration. Several studies have suggested that flavonoids can alleviate learning and memory deficits by targeting this signaling pathway [29,203,204,205]. However, other pathways, including the MAPK-ERK1/2 cascade [206], have also been proposed and outlined in a recent review [207]. In this section, we will first explore the impact of Aβ- and tau-mediated neuroinflammation on synaptic plasticity-related neuronal signaling. We will support the necessity of the PI3K/AKT/GSK-3β pathway in AD treatments and investigate the potential roles of kaempferol and quercetin in improving memory and cognition through this pathway. 

### 5.1. Neuroplasticity Deficits in AD

An ideal AD treatment should enhance the expression of plasticity-related genes such as BDNF, a neurotrophic factor that regulates neuronal plasticity and survival [208,209,210,211,212,213,214]. BDNF signaling begins with its binding to the receptor, Trkβ, activating signaling via a variety of pathways like PI3K/AKT [211,215]. Then, AKT or protein kinase B (PKB) [216] can activate the CREB-mediated transcription of BDNF [217,218]. Since Trkβ receptors mediate the pro-neuroplastic effects of BDNF [219], AD drugs must produce a direct or indirect effect on the receptor. 

BDNF deficits increase the risk of AD development [220], and BDNF dysfunction due to impaired PI3K and AKT signaling can expedite neurodegeneration [7,41,221,222,223]. The PI3K/AKT signaling pathway has multiple functions, including regulating synaptic plasticity, glucose processing, cell cycle progression, cell proliferation, survival, and apoptosis [167,175,224,225,226]. Moreover, this pathway may protect neurons from Aβ toxicity [224], oxidative stress [227], and neuroinflammation [217]. GSK-3β is downstream of PI3K/AKT, and Aβ can specifically lead to its hyperactivity [7]. However, BDNF and CREB are also vulnerable to Aβ signaling [228] as CREB is regulated by the PI3K/AKT/GSK-3β pathway [211,212,215,229,230,231].

Thus, the Aβ-mediated signaling cascade that degenerates the neuron is as follows (Figure 4A): Aβ binding to NMDAR inhibits PI3K/AKT signaling by activating GSK-3β-mediated tau hyperphosphorylation and CREB downregulation [93,97,210,211,223,229,230,232,233,234,235,236,237]. Consequently, the impaired CREB-mediated transcription of BDNF genes decreases plasticity and facilitates plaque accumulation, as demonstrated in postmortem tissues from humans and human neuronal cells [209,210,229,232]. The absence of protective BDNF and PI3K/AKT activity facilitates the caspase-mediated pro-apoptotic signaling cascade [6,224], degenerating the neuronal circuitry, while tau dissociation from microtubules breaks down the neuronal cytoskeleton [7,233,238,239,240,241,242]. 

However, future AD treatments could reverse this toxic signaling via the following mechanism: A drug must either directly activate Trkβ or should do so indirectly by enhancing BDNF transcription [210]. The drug can either directly activate PI3K and/or AKT, which would ultimately inhibit GSK-3β via the phosphorylation of its Ser9 residue [224]. In turn, AKT can also inhibit caspase-9 and Bcl-3 to inhibit pro-apoptotic signaling [243,244,245,246]. One study showed that the GSK-3β inhibitor, AR-A014418 (ARA), inhibited BACE1-mediated APP cleavage into Aβ proteins in rodents [48], supporting the necessity of a GSK-3β-inhibiting drug for the treatment of AD. Finally, GSK-3β inhibition also reversed oxidative stress [93,247]. In short, the PI3K/AKT pathway can not only reverse neuroinflammation but can also counteract Aβ-mediated tau hyperphosphorylation by inhibiting GSK-3β. 

### 5.2. Quercetin and Kaempferol Resolve AD-Related Plasticity Deficits

The multimodal mechanisms of kaempferol and quercetin collectively slow neurodegeneration by combating the impairments that are illustrated in Figure 4A and are described in Table 1. Specifically, the restoration of proper PI3K/AKT signaling will greatly improve synaptic plasticity deficits in AD [7]. While quercetin’s interaction with each component of this signaling pathway has already been documented [7], kaempferol’s mechanisms are still unclear. However, since kaempferol’s structure is similar to that of quercetin [165], we propose that kaempferol has a nearly identical mechanism with respect to the signaling pathway in this subsection. Finally, we will propose the potential outcomes of these molecular interactions.

Molecular docking studies suggested that quercetin can bind PI3K, AKT, and GSK3β [213,250,255,256,257,260,264,265]. Specifically, quercetin can bind to PI3K [256], consequently activating AKT signaling [265], or quercetin can directly bind to AKT [257]. In preclinical studies, quercetin reduced GSK-3β activity, which decreased tau hyperphosphorylation and reduced pro-apoptotic signaling [7,38,159]. Quercetin treatment in rodents also increased BDNF, Trkβ, PI3K, and AKT expression [243,266]. Consequently, quercetin enhanced neurite outgrowth in hippocampal neurons [36] and ameliorated the stress-induced downregulation of CREB and BDNF [40], suggesting that quercetin could potently replenish neuroplasticity in the AD brain. Moreover, quercetin inhibited Aβ by restoring Trkβ signaling and CREB-mediated BDNF transcription, increasing the viability of SH-SY5Y cells [252]. Finally, quercetin’s dual pro-neuroplastic and anti-inflammatory effects may also be related to the quercetin-mediated downregulation of BACE1 expression via the inhibition of NF-kB [253,254,264,267]. Taken together, these data suggest that quercetin antagonizes Aβ-induced GSK-3β signaling relative to tau by activating the PI3K/AKT pathway and directly inhibiting GSK-3β [7,225,241,255,256,260]. Consequently, proper BDNF levels can be restored to replenish neuronal plasticity in the AD brain. Similar chemicals, such as epigallocatechin-3-gallate (EGCG), attenuated tau hyperphosphorylation in a similar mechanism [23,268,269,270]. Thus, quercetin clearly has dual neuroprotective and pro-neuroplastic mechanisms in cells [33,65,252], and the clinical outcomes of quercetin’s pro-neuroplastic mechanisms were supported by its memory and cognition-boosting effects in rodent models of AD and Parkinson’s disease [23,38,44,271,272,273,274,275,276]. Select molecular targets of quercetin are described in Table 1.

Kaempferol may have similar pro-neuroplastic mechanisms to quercetin, and some of its molecular targets are outlined in Table 1. First, kaempferol improved hippocampal plasticity following traumatic brain injury in young rodents [277] and improved memory in rodents [29,57] and Drosophila [173]. Moreover, kaempferol dose-dependently maintained cell viability following Aβ treatment in multiple studies [29,149,195,248]. This could be due to kaempferol’s inhibition of BACE1-mediated Aβ synthesis [253,254] or the activation of the PI3K/AKT signaling pathway, enhancing CREB-mediated BDNF transcription [175,211,258]. Although one molecular docking study suggested that kaempferol may have minimal affinity for GSK-3β [250], kaempferol likely inhibits GSK-3β indirectly by first binding and activating PI3K [256] or AKT [175,185,257]. Via this mechanism, kaempferol prevents tau hyperphosphorylation, protecting neuronal morphology and function [47,278,279,280,281]. Then, AKT can activate CREB-mediated BDNF transcription [217]. Supporting this pro-neuroplastic mechanism, kaempferol and its metabolite, kaempferide, produced similar effects that resulted in Trkβ signaling [171,210] and enhanced BDNF expression in Aβ-treated mice [243]. Taken together, these data suggest that kaempferol enhances neuroplasticity to reverse Aβ damage by activating the PI3K/AKT cascade, which potentiates CREB-mediated BDNF transcription. However, kaempferol produces the opposite effect on this signaling pathway in microglial cells [167] and cancer cells [282]. Thus, kaempferol’s effects on the PI3K/AKT signaling cascade are dynamic and depend on cell lineage. 

Despite the lack of literature demonstrating a direct modulation of tau by kaempferol, there is plenty of evidence to support the possibility that kaempferol inhibits tau hyperphosphorylation via the PI3K/AKT pathway and by antagonizing Aβ-mediated GSK-3β signaling [29,149,195]. This mechanism prevents neuronal degeneration and a loss of synaptic plasticity. Thus, the pro-neuroplastic effect of kaempferol requires the inhibition of GSK-3β and CREB phosphorylation. Remarkably, a recent molecular docking study suggested that kaempferol could bind to NMDAR [259]. However, in vivo studies are still required to confirm this effect.

These data suggest a clear anti-AD mechanism of quercetin and kaempferol, as outlined in Figure 4B. First, quercetin and kaempferol could enter the cell cytoplasm due to their lipophilic polyphenolic structure. Quercetin and kaempferol scavenge ROS and activate PI3K/AKT signaling to inhibit GSK-3β. Specifically, they can bind directly to PI3K or AKT to activate protective signaling, inhibiting GSK-3β and preventing tau hyperphosphorylation. This signaling cascade reduces the formation of NFTs in the AD brain. GSK-3β inhibition can also antagonize Aβ-NMDAR interactions. Thus, downstream pro-apoptotic signaling mediators are also inhibited by quercetin and kaempferol treatment. Due to reduced NFT and amyloid plaque formation, microglial hyperactivity decreases in the absence of the burden of clearance. Thus, progressive neuroinflammatory signaling is slowed, allowing surrounding neuronal synapses to survive. After chronic quercetin treatment, progressive elevations in BDNF release rebuild damaged synapses by favoring neurotrophic signaling over cytotoxic Aβ signaling, improving memory and cognition. Of note, molecular docking studies have not supported the possibility that kaempferol and quercetin can directly bind to tau protein, supporting their indirect inhibitory mechanism via GSK-3β inhibition. Taken together, kaempferol and quercetin share multiple mechanisms that slow AD progression by first limiting ROS activity, NFT aggregation, and Aβ-mediated toxic signaling, slowing neurodegeneration. 

**Figure 4 biology-12-01453-f004:**
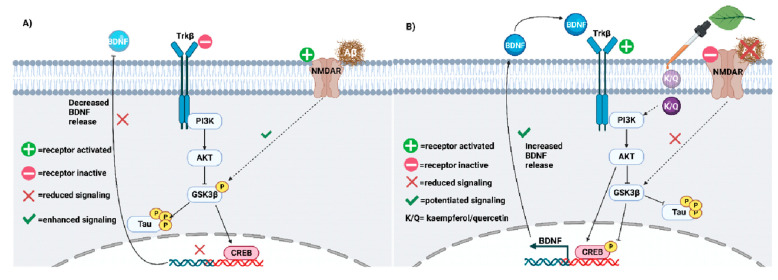
(**A**) Neuroplasticity deficits accelerate AD progression and must be treated. Impaired PI3K-AKT signaling facilitates GSK3β-mediated phosphorylation of tau. Aβ may potentiate tau hyperphosphorylation via GSK3β. (**B**) Kaempferol and quercetin (K/Q) invoke the PI3K/AKT pathway to antagonize Aβ and reduce tau hyperphosphorylation in neurons. As a result, neuroplasticity is increased in the AD brain [283].

## 6. Quercetin and Kaempferol in Common Herbs

Although data on the co-treatment of quercetin and kaempferol are still somewhat limited, the abundance of both compounds in several common herbs requires the investigation of the synergistic effects of both flavonoids, in addition to their interactions with other herbal phytochemicals. Flavonoid-rich herbs are commonly employed in traditional Chinese medicine (TCM), in which an emphasis is placed on the utility of natural treatments. Moreover, these herbs are generally safe for consumption [224]. Kaempferol is the second most common flavonoid in traditional medicinal herbs, following quercetin [225,284]. Other reviews have assessed the efficacy and safety of natural medicine in the treatment of neurodegenerative diseases [7,224], highlighting the potential medicinal properties of herbs in treating AD. Flavonoids are commonly found in herbs such as *Schima wallichii* Korth, *Maesa membranacea*, *Ginkgo biloba*, and many more [175,225,278]. These phytochemicals could work synergistically with each other and with other herbal components to invoke anti-AD effects. Thus, we explore common herbal sources of kaempferol and quercetin, describe the anti-AD mechanisms of herbs, and propose a design for a future AD treatment based on the current evidence of these effects.

*Ginkgo biloba* is a quercetin- and kaempferol-rich herb proposed to treat AD [285]. *G. biloba* improves memory and cognition by inhibiting ROS, facilitating hippocampal neuron proliferation, halting Aβ plaque accumulation, and reducing tau hyperphosphorylation [47,286,287,288]. Moreover, this effect is associated with reduced GSK-3β activity and the increased expression of PSD-95 and synapsin-1 [47]. As seen with kaempferol and quercetin alone, *G. biloba* potentiates PI3K/AKT relative to CREB signaling to promote neuroplasticity [287,289,290,291,292,293]. Hippophae rhamnoides extracts are also rich in quercetin and kaempferol, and they enhanced neuronal differentiation and neurite outgrowth via PI3K/AKT and ERK signaling [294,295]. However, clinical trials have revealed the inconsistent efficacy of *G. biloba* on cognition and other AD-related parameters [296]. Camellia sinensis is another kaempferol- and quercetin-rich herb commonly grown to produce black and green tea [297,298]. *C. sinensis* extracts improved spatial memory and reduced hippocampal Aβ fibrillization in AD rodents and had greater antioxidant effects compared to other herbs [298,299]. Kaempferol and its derivatives are found in the leaves of *Maesa membranacea*, *Schima wallichii* Korth, *Carthamus tinctorius*, *Panax ginseng*, and several other herbs [175,188,225,278]. *S. wallichii* was neuroprotective due to the promotion of hippocampal and cortical AKT signaling [175], and *M. membranacea* could protect H202-treated SH-SY5Y cells [225] and hippocampal tissue [300] via the same pathway due to their kaempferol abundance. C. tinctorus is rich in kaempferol, produces a similar effect, and invokes protective AMPK signaling [188]. Finally, recent studies also suggested that other herbs such as *Morenga oleifera*, *Cuscuta chinensis*, *Allium cepa*, *Litchi chinensis*, *Prakia roxburghii*, *Radix astragali*, *Acoritatan Fagopyrum tataricum*, *Carthami flos*, *Punica granatum*, and *Cyperi rhizoma* [251,257,264,301,302,303,304,305] may also be great sources of kaempferol and/or quercetin and produce anti-AD effects. Their medicinal properties and expression of kaempferol and quercetin are outlined in Table 2.

Polyherbal cocktails, such as Chaihu shugan san (CSS) and Huangqi Sijunzi (HQSJDZ), could treat AD and its risk factors. CSS is abundant in kaempferol and quercetin and contains herbs such as *Glycyrrhiza uralensis*, *Cyperus rotundus*, and *Buplerum falcatum* [256]. Specifically, the antidepressant effect of CSS is mediated by increased PI3K/AKT/BDNF signaling and decreased GSK-3β and IL-2 activity [256], suggesting that polyherbal cocktails may be protected from AD development. HQSJDZ, rich in kaempferol and quercetin, had cholinergic, anti-inflammatory, and anti-GSK-3β effects [278,306]. Moreover, a cocktail of *C. sinensis*, Hypericum perforatum, and Bacopa monnieri produced robust antioxidant effects compared to single-herb treatment [298]. These data suggest that polyherbal treatment may be superior to single-herb therapy. 

Due to the well-documented effects of quercetin and kaempferol on Aβ, GSK-3β, PI3K/AKT, and multiple pro-inflammatory molecules, it is possible that both phytochemicals, given their abundance, contribute vastly to the anti-AD effects of several herbs. Such herbs include *Ginkgo biloba*, *Camellia sinensis*, *Glycyrrhiza uralensis*, *Cyperus rotundus*, and *Buplerum falcatum*. The herbal sources outlined in Table 2 may also be great additions to the treatment protocol that can enhance the dietary intake of kaempferol and quercetin. According to the practice of TCM, it is possible that a multi-herb cocktail containing varying amounts of these herbs could alleviate AD symptoms, as seen with current medications, but it may also halt progression relative to a unique multi-modal mechanism. Multiple studies have suggested that the synergistic effects of polyherbal treatments produce greater anti-AD efficacy compared to single-herb treatment [256,278,298]. Thus, the research and development of future AD drugs should consider the applications of these common herbs in future drug cocktails. On the other hand, since clinical trials featuring *Ginkgo biloba* extracts have demonstrated controversial results on the progression of AD [296], single-herb treatments may be insufficient to treat AD.

**Table 2 biology-12-01453-t002:** Plant sources of kaempferol and quercetin and/or their metabolites and a description of reported herbal health effects.

Species Name	Kaempferol	Quercetin	Example Health Effects	Reference
*Ginkgo biloba*	+	+	Memory and cognition improvement	[285,296,307,308]
*Camellia sinensis*	+	+	Improved memory and antioxidant effects	[297,298,299]
*Maesa membranacea*	+	+	Neuroprotective	[175,188,225,278]
*Schima wallichii* Korth	+	−	Neuroprotective	[175,187,225,278]
*Carthamus tinctorius*	+	+	Neuroprotective	[175,187,225,278,309]
*Panax ginseng*	+	+	Neuroprotective	[175,187,225,278,310]
*Morenga oleifera*	+	+	Memory improvement	[300,301,302]
*Cuscuta chinensis*	+	+	Memory improving, Neuroprotective, Hepatoprotective, Immunomodulatory	[311]
*Allium cepa*	+	+	Anti-inflammatory	[312,313]
*Hippophae rhamnoides* L.	+	+	Anti-inflammatory	[294,295]
*Litchi chinensis*	+	+	Neuroprotective	[303,314]
*Prakia roxburghii*	-	+	Neuroprotective	[304]
*Radix astragali*	+	+	Neuroprotective	[213]
*Fagopyrum tataricum* (L.)	+	+	Decrease neurotoxicity	[251]
*Carthami flos*	+	+	Anti-ischemic	[213]
*Punica granatum*	+	+	Anti-inflammatory	[264,315]
*Cyperi rhizoma*	+	+	Antidepressant	[257]

## 7. Limitations of Kaempferol and Quercetin Treatment 

### 7.1. Bioavailability

Despite the promising effects of these herbs and flavonoids in AD treatment, low bioavailability and blood-brain barrier (BBB) permeability are common obstacles interfering with drug delivery to the brain [316,317,318]. Thus, structural manipulations are commonly required to improve the bioavailability of flavonoids. Moreover, the varying dietary intake of macromolecules like fats and carbohydrates also impacts BBB permeability relative to polyphenols [44]. Other factors, such as aging or diagnosis with AD, may increase BBB permeability to peripheral chemicals [39,319,320,321]. However, tau hyperphosphorylation and astrocytic hyperactivity invoke neuroinflammatory signaling that damages BBB integrity and increases its permeability [112,322,323,324]. The limited BBB permeability may also explain the lack of clinical trials in humans [68].

Despite its lipophilicity and easy oral administration in common foods, quercetin treatment for AD may be challenged by its limited bioavailability relative to the brain [3,258]. Since quercetin absorption is predominantly mediated by the small intestine, it is vulnerable to extensive first-pass metabolism [133,258,325]. While its distribution was evidenced in the plasma, liver, heart, spleen, kidneys, and lungs, quercetin levels were non-detectable in the rat brain [326,327]. Hence, it has around 65% BBB permeability [321,328] and is absorbed in the stomach with 24–53% bioavailability [329]. The P-glycoprotein transporter, which is a BBB efflux transporter, has a high affinity for free, unaltered quercetin and greatly reduces its bioavailability by pumping quercetin away from the brain [51,330]. While in vitro studies showed the promising antioxidant effects of quercetin, most studies in animal models have demonstrated limited efficacy [3,331]. These data show that quercetin’s limited bioavailability could debilitate anti-AD effects [258].

Chemical modifications are necessary to ensure quercetin distribution to the brain, as some metabolites may also have higher efficacy than quercetin alone. For instance, quercetin–glucoside conjugation enhanced its bioavailability [129]. Quercetin glycosides are commonly available in fruits and vegetables, improving its delivery to the CNS [51,332]. Glucuronidation in the liver also increased the distribution of quercetin to the brain in oxidative stress models [23]. Moreover, in vivo studies showed that lipid nanoparticle-loaded quercetin enhances its entry into the brain [39,44,146,158,163,333,334]. Moreover, quercetin loading into selenium nanoparticles improved brain distribution and anti-Aβ mechanisms [335]. However, excess selenium levels in the body can produce oxidative stress [336,337], potentially limiting the clinical efficacy of this approach.

Like quercetin, free kaempferol generally has low oral bioavailability due to metabolic degradation [324,338,339]. Kaempferol is generally slowly absorbed in the GI tract and can be distributed to several tissues [326,340], suggesting that the primary limitation of kaempferol treatment is limited bioavailability. However, several modifications to improve its BBB permeability have been proposed. First, nanoparticle loading also improves kaempferol bioavailability [194,334,341,342,343,344], and kaempferol–sugar conjugates also demonstrate superior protective efficacy [36]. For instance, nanoparticle-loaded kaempferol has more robust anti-inflammatory effects than kaempferol alone [68]. Clinical trials revealed that quercetin had superior memory-modulating activity in AD patients compared to healthy elderly controls [345,346,347], suggesting that the increased BBB permeability in AD may, in turn, improve flavonoid bioavailability and efficacy in neurodegenerative brains. Several other forms of delivery have been proposed for both flavonoids, including gold-infused nanoparticles [348,349], multi-targeted drugs [350], extracellular vesicles [351], and intranasal administration [352]. Finally, other proposed nanoformulation delivery systems include nanomatrixes, nanoemulsions, nanostructured lipid carriers, and nanocomplexes [343,344,353].

### 7.2. Adverse Health Effects and Other Limitations

Most studies show promising medical benefits for kaempferol and quercetin and suggest that they are safe in a variety of doses. For example, quercetin is included in the Food and Drug Administration’s Generally Recognized as Safe (GRAS) list for supplemental use of up to 500 mg per serving in foods and beverages [129,354]. However, flavonoids’ clinical efficacy may also be limited by adverse effects [329]. While the Ames test suggested that quercetin could have carcinogenic properties, most studies have opposed this finding and suggested that quercetin is safe [355]. One study suggested that high-dose quercetin treatment reduced neuronal survival, induced oxidative stress, and inhibited AKT [356]. Thus, physicians should carefully manage the abundance of quercetin in the AD patient’s diet to maintain its proper anti-degenerative effects. Moreover, the efficacy of quercetin may be limited in AD patients who are also diagnosed with leukemia, as quercetin inhibits the PI3K/AKT signaling pathway in HG3 cells [282]. It is possible that, since most dietary quercetin is distributed to peripheral sites, lower concentrations in the brain may decrease its efficacy in AD.

Although kaempferol is most likely safe to consume [339] and most studies showed low toxicity in mice [357,358,359], some studies have reported concerns about potential mutagenic effects in people with iron and folic acid deficiencies [338,339,360]. Since the excess inhibition of GSK-3β may produce toxic effects in cells [233], kaempferol’s low-affinity GSK-3β interactions may underlie its generally low toxicity. In a 4-week randomized, double-blind clinical trial, participants were divided into a group that received 50 mg of kaempferol daily and a placebo group; kaempferol was reported as mostly safe, but the small sample size of 24 in each group limits this study [359]. Overall, the majority of work on the herb suggests it to be safe, even in high doses, but more clinical trials are highly recommended.

## 8. Discussion

Since AD still lacks a true cure, and currently available medications are insufficient to halt disease progression, the field has sought out multimodal treatments for AD. However, little progress in drug development has been made in recent decades, necessitating new alternative treatments. Thus, the objective of this review was to deduce the anti-AD mechanisms of kaempferol and quercetin. These phytochemicals were selected for multiple reasons, including their abundance [38,116] and their multimodal mechanisms (Figure 5) that include antioxidant, anti-inflammatory, pro-neuroplastic, and neuroprotective effects. Thus, quercetin and kaempferol may treat Alzheimer’s disease, and we aimed to explore their anti-amyloidogenic, antioxidant, anti-inflammatory, anti-tau, and pro-neuroplastic mechanisms [6,29,38,39,51,127,128,149,159,167,361]. In turn, phytochemicals may not only reduce AD symptoms [29,33,132] but also delay the progression of the disorder. Of note, the efficacy of these flavonoids to produce the effects outlined in this review depends on any chemical modifications that may occur throughout the absorption and distribution of phytochemicals to the brain.

Perhaps the most significant contribution of this review is the complex anti-degenerative mechanism of kaempferol. We utilized the available literature to show that kaempferol’s dual anti-tau and anti-Aβ mechanisms are due to its modulation of the PI3K/AKT/GSK-3β signaling pathway. Both phytochemicals resolve oxidative stress by increasing antioxidant levels and inhibiting ROS signaling [119]. Meanwhile, they halt inflammatory signaling [29,38] to commence a neuroprotective effect. Then, resolved microglial and astrocytic activity facilitates proper Aβ clearance from the brain [6] and reduces continued neuronal damage due to the neuroinflammatory environment [122,188,195]. The modulation of PI3K/AKT/GSK-3β and Trkβ/BDNF signaling potentiates neuroplasticity and protects neurons from insults like Aβ [10,240], decreasing tau hyperphosphorylation and preserving the neuronal cytoskeletal structure. These phytochemicals, in turn, protect neuronal networks [33,40], improving memory and cognitive function in AD patients. Other flavonoids with heterocyclic structures [362], including morin [363,364,365,366], rutin [367,368], and luteolin [369,370,371], share many similar anti-AD properties relative to kaempferol and quercetin. However, rutin [368] failed to increase BDNF levels, like kaempferol and quercetin.

Due to the superior efficacy of polyherbal treatments, such as HQSJDZ and CSS [256,278,298], we proposed that polyherbal treatment, containing quercetin- and kaempferol-rich herbs like *Ginkgo biloba*, *Camellia sinensis*, *Glycyrrhiza uralensis*, *Cyperus rotundus*, and *Buplerum falcatum* may produce superior anti-AD efficacy compared to single-herb supplements. Recent studies also suggested that herbs such as Morenga oleifera, Cuscuta chinensis, Allium cepa, Hippophae rhamnoides, Litchi chinensis, Prakia roxburghii, Radix astragali, Fagopyrum tataricum, and Carthami flos [251,294,301,302,303,304] may also be candidates for polyherbal treatment. However, a recent review noted that kaempferol and quercetin are widely available in hundreds of herbs, and it is possible that they may not be as abundant as other phytochemicals in some species [372], supporting the necessity of polyherbal treatment to obtain biologically effective concentrations.

As previously mentioned, clinical trials suggest that kaempferol and quercetin could treat AD in humans [135,346,347,373,374], but single-herb treatment was unsuccessful in clinical trials [296]. Future trials should assess bioavailability-enhancing delivery methods for quercetin and kaempferol. However, recent studies also suggested that both quercetin and kaempferol have the ability to maintain and protect BBB integrity [375,376,377,378,379]. This could possibly be due to their anti-inflammatory properties that could be invoked if they reach the brain. Of course, clinical trials should continue to assess the efficacy of herbal sources in AD-related symptoms. However, the misuse of herbal treatments may produce side effects, including gastrointestinal discomfort, insomnia, and tachycardia [298]. Thus, studies assessing these side effects are limited and require further investigation [36,37]. Nonetheless, these natural herbs are generally considered safe, and toxic effects are uncommon [51,116,142]. Finally, an investigation of interactions between these polyphenols and other drugs commonly prescribed to AD patients is required.

Although the data presented in this review showcase the great potential of these herbs in AD treatment, a few limitations have impacted this review. Specifically, studies investigating the tau hyperphosphorylation-inhibiting mechanisms of these herbs may be limited due to the rapid dephosphorylation of the protein in postmortem AD tissues [15,279]. Moreover, the abundantly described bioavailability limitations of both herbs critically limit the efficiency of human studies. This could be one reason underlying the lack of kaempferol and quercetin’s clinical efficacy to date. Clinical trials investigating compounds that increase the bioavailability of these phytochemicals are still needed. Since quercetin and kaempferol are naturally abundant in the average diet, future clinical trials can be easily conducted. Finally, while molecular docking studies show the potential pharmacodynamic interactions between kaempferol/quercetin and the outlined pro-neuroplastic targets, these approaches are merely estimates of binding affinity based on the crystal structures of the target protein and the molecular structures of the ligand, and they could be vulnerable to mispredictions [380]. Thus, future studies must either employ competition assays or ligand inhibitor/antagonist studies to confidently elucidate the true affinity of kaempferol and quercetin for the targets of interest. Nonetheless, recent data support the exciting potential of kaempferol and quercetin to slow the progression of AD and alleviate the symptoms.

## 9. Conclusions

Kaempferol and quercetin clearly exhibit multimodal mechanisms that halt AD progression and alleviate symptoms. Given the multifaceted nature of AD pathogenesis, future treatments need to adopt a multimodal approach that targets the Aβ-tau signaling pathway via the modulation of the PI3K/AKT/GSK3β signaling cascade, leading to a pro-neuroplastic effect via enhanced BDNF signaling. To our knowledge, our review demonstrates how kaempferol and quercetin address various aspects of AD, including neuroinflammation, oxidative stress, reduced plasticity, and Aβ and tau signaling. Notably, our review is the first to propose that kaempferol can mitigate both tau hyperphosphorylation and Aβ toxicity by directly targeting the PI3K/AKT/GSK3β pathway. Additionally, we suggest that polyherbal cocktails rich in kaempferol and quercetin may yield robust anti-AD effects, and we identified potential herbal sources of kaempferol and quercetin. Finally, we discuss the limitations that currently impede the efficacy of kaempferol/quercetin treatment, and suggest potential adjustments to circumvent these challenges. Together, these changes can improve the anti-AD efficacy of natural flavonoids and could be ideal adjunctive or alternative treatments to currently available drugs.

## Figures and Tables

**Figure 1 biology-12-01453-f001:**
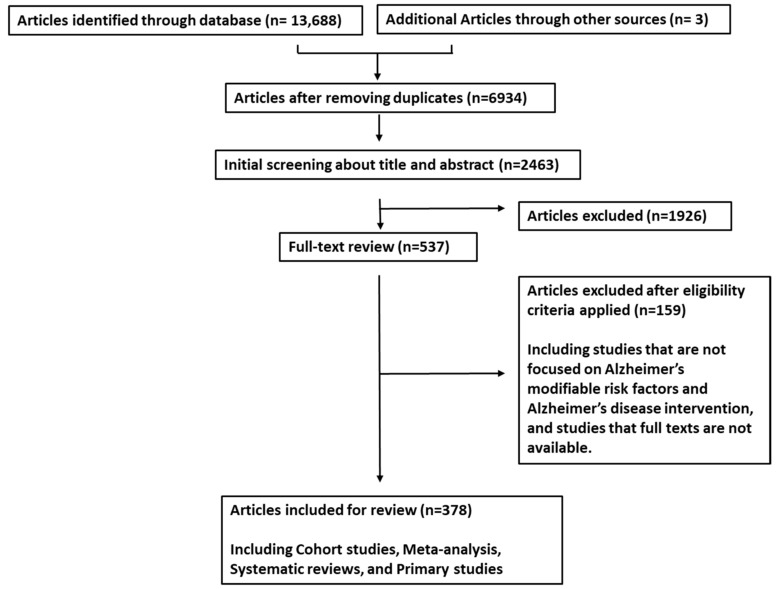
Flowchart depicting the article screening and selection process according to PRISMA guidelines.

**Figure 5 biology-12-01453-f005:**
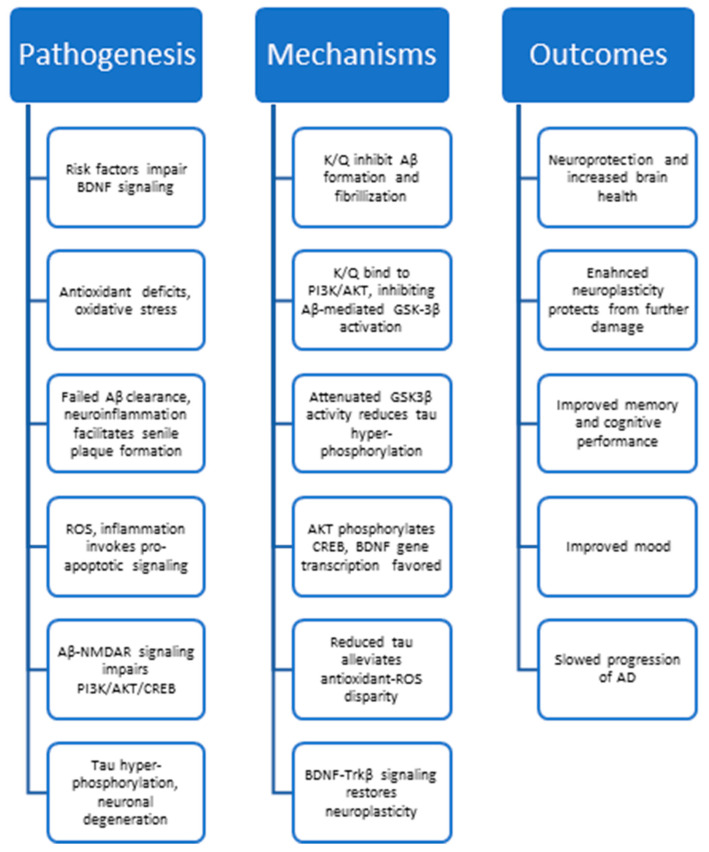
A graphical summary of the underlying mechanisms behind AD progression (pathogenesis), the proposed mechanisms of kaempferol and quercetin (K/Q), where K/Q represents kaempferol and quercetin, and the impact of these molecular changes on behavior and disease progression (outcomes). Each category is presented in a top-to-bottom chronological order.

**Table 1 biology-12-01453-t001:** Kaempferol and Quercetin and molecular interactions with select molecules relevant to neuroplasticity in AD. These affinity or potency values are deduced from molecular docking studies (affinity) and competition assays (IC50; potency) or were indirect interactions evidenced in the literature. Docking scores (DS) of 5 or higher indicate the high affinity of a compound for the protein of interest [248,249]. Or, affinity from docking studies may be expressed as binding energies (BE) in -kcal/mol. The more negative the value, the higher the binding affinity. If studies have not supported direct binding to a certain target, the affinity column is noted as “Indirect”.

Molecular Target	Phytochemical	Mechanism	Affinity (DS, BE, or IC50)	References
GSK-3β	Kaempferol	Inhibit	4.6 (DS, mice); −7.9 kcal/mol (human brain docking) −9.2 kcal/mol (zebrafish)	[243,250,251]
	Quercetin	Inhibit	5.64 (DS); −8.8 kcal/mol (human brain docking) −9.0 kcal/mol (zebrafish)	[243,250,251]
Aβ	Kaempferol	Inhibit	Indirect	[171]
	Quercetin	Inhibit	Indirect	[252]
BACE1	Kaempferol	Inhibit	IC50 = 14.7 µM	[253,254]
	Quercetin	Inhibit	IC50 = 5.4 µM	[253,254]
Tau	Kaempferol	Inhibit hyperactivation	Indirect	[47]
	Quercetin	Inhibit hyperactivation	Indirect	[255]
PI3K	Kaempferol	Activate	5.19 (DS, neurons)	[256]
	Quercetin	Activate	7.04 (MD, neurons)	[256]
AKT1	Kaempferol	Activate	5.13 (MD, neurons); −9.3 kcal/mol	[256,257]
	Quercetin	Activate	5.03 (MD, neurons), −9.4 kcal/mol; −7.96 kcal/mol	[213,256,257]
BDNF	Kaempferol	Upregulate	Indirect	[258]
	Quercetin	Upregulate	Indirect	[252]
CREB	Kaempferol	Activate	Indirect	[211]
	Quercetin	Activate	Indirect	[252]
NMDAR	Kaempferol	Reverse Aβ binding	−10.84 kcal/mol	[259]
	Quercetin	Reverse Aβ binding	Indirect	[255,260]
HDAC	Kaempferol	Activate	Not Found	[188,189]
	Quercetin	Activate	IC50 = 105.1 µM	[261]
AChE	Kaempferol	Inhibit	−10.26 kcal/mol; between −8.6 and −9.22 kcal/mol	[259,262,263]
	Quercetin	Inhibit	−7.9 kcal/mol; IC50 = 4.59 ± 0.27 µM	[155,263]

## Data Availability

Not applicable.

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
