# Peer review of "Polyherbal and Multimodal Treatments: Kaempferol- and Quercetin-Rich Herbs Alleviate Symptoms of Alzheimer’s Disease"

_biology, 2023, doi:10.3390/biology12111453_

Round 1
Reviewer 1 Report
Comments and Suggestions for Authors Recommendations for improving the manuscript Brief summary is very long. It should be one short paragraph with aim of paper and main contributions and strengths. Proposed simple summary should be made shorter and clearer. Graphical abstract of the underlying mechanisms behind AD pathogenesis, proposed mechanisms of kaempferol and quercetin is very informative and is an original idea. But I recommend it will be integrated in the body of Discussion in order to confirm the key conception of this review. Only 2 figures were included in the review. I recommend more illustration to be added (for example some tables- about plants containing both flavonoids etc.), as well as the chemical structures of the two bioflavonoids, demonstrating their similarities. Figure 2B (as well as 2A) mentioned in the text about the anti-AD mechanisms of quercetin and kaempferol are missing in the manuscript. Bibliography should be carefully corrected according the requirements of the Journal. The references contains 273 authors, but less than 40 of them are from the last 5 years (i.e. only 15 %). It is necessary to increase % of papers cited in the last 5 years. I recommend careful correction of the text containing several mistakes in numbering or typing mistakes as well as some repetition of words noticed in the manuscript. There are also missing explanations of some abbreviations in the text (for example for acetylcholine).
Author Response
Reviewer 1 Comments:
Recommendations for improving the manuscript Brief summary is very long. It should be one short paragraph with aim of paper and main contributions and strengths. Proposed simple summary should be made shorter and clearer. Graphical abstract of the underlying mechanisms behind AD pathogenesis, proposed mechanisms of kaempferol and quercetin is very informative and is an original idea. But I recommend it will be integrated in the body of Discussion in order to confirm the key conception of this review. Only 2 figures were included in the review. I recommend more illustration to be added (for example some tables- about plants containing both flavonoids etc.), as well as the chemical structures of the two bioflavonoids, demonstrating their similarities. Figure 2B (as well as 2A) mentioned in the text about the anti-AD mechanisms of quercetin and kaempferol are missing in the manuscript. Bibliography should be carefully corrected according the requirements of the Journal. The references contains 273 authors, but less than 40 of them are from the last 5 years (i.e. only 15 %). It is necessary to increase % of papers cited in the last 5 years. I recommend careful correction of the text containing several mistakes in numbering or typing mistakes as well as some repetition of words noticed in the manuscript. There are also missing explanations of some abbreviations in the text (for example for acetylcholine).
Response:
We greatly appreciate your recommendations and have incorporated them into the manuscript. We sought out as many recent papers (from the past 5 years) as we could within this given time. We found over 100 new articles that were critical to the improvement of the manuscript. However, we have still kept many of the older references, as we felt that these references were critical to the reader’s understanding of the literature. Moreover, many of these older references were deduced from articles that we had been reading as our primary references and some of these articles were so intriguing to us that we decided to read these older papers and incorporate them into the manuscript. Notably, in order to satisfy the requests of a some of the other reviewers, we have added mostly new references, but may have also added a few older references. We feel that it is critical to ensure that the original work that we find in our newer readings is properly given the credit they deserve. We have also followed your recommendation to move the flowchart summary into the discussion section, and we feel that it fits in greatly with the discussion to wrap up the main ideas of the article. We also have noticed the repetitive nature of some of the writing, and we have thoroughly read through and corrected many of these repetitive sentences and have consolidated some of the material to alleviate the grammatical errors. We believe that this has greatly improved the manuscript. These changes should make it easier for the readers to follow, especially with section 5.1. With that section, we noticed that the repetitive nature of some of our writing may have interfered with the reader’s ability to follow the main points, so we consolidated this information and organized it accordingly. We believe that this follows the mechanism outlined in Figure 4 accordingly. Moreover, we loved your suggestion about incorporating some of the potential plant species that are sources of kaempferol and quercetin into a table, and we have made a new table (Table 1) accordingly. This table will be very useful to readers seeking more information about traditional medicinal sources of quercetin and kaempferol. There is also an abundance of new literature that has identified these phytochemicals in these new species. We did not significantly describe these new species in the body of the text, as we felt that it would lead to repetitive wording and could potentially result in a loss of the reader’s interest. However, we point out the table as a source for the latest sources of kaempferol and quercetin in case readers want to explore these sources further. We have also incorporated a table demonstrating the potency or affinity of kaempferol and quercetin for select molecular targets (Table 2). Next, we followed your advice to add images of the chemical structures of quercetin and kaempferol, in sections 4.1 and 4.2, respectively. We think that this was a great improvement to the manuscript and provides the reader with visual evidence of the structural similarities of the two flavonoids. Finally, we have generated a list of abbreviations from this paper, and we will send it alongside this revised manuscript.

Reviewer 2 Report
Comments and Suggestions for Authors
The manuscript titled "Polyherbal and Multimodal: Kaempferol- and Quercetin-Rich Herbs Alleviate Symptoms of Alzheimer’s Disease" provides a comprehensive overview of the potential therapeutic effects of kaempferol and quercetin in the treatment of Alzheimer's Disease (AD). The paper follows the PRISMA guidelines for a systematic review and highlights the importance of a multimodal approach to address the complex pathogenesis of AD.
The manuscript critically analyzes various aspects of the pathophysiology of AD, including the role of Aβ plaques, tau hyperphosphorylation, oxidative stress, neuroinflammation, and impaired synaptic plasticity. It effectively demonstrates how kaempferol and quercetin, two phytochemicals commonly found in various herbs, exhibit promising anti-AD effects through their antioxidant, anti-inflammatory, and neuroprotective properties.
The strengths of this manuscript lie in its comprehensive approach to examining the mechanisms of action of kaempferol and quercetin, emphasizing their potential to alleviate multiple aspects of AD progression. The inclusion of relevant data and studies enhances the credibility of the proposed mechanisms and therapeutic potential of these phytochemicals.
However, several notable limitations need consideration. While the manuscript extensively discusses the potential benefits of kaempferol and quercetin, it lacks a thorough discussion of the challenges associated with the translation of these findings into effective clinical treatments. The issue of limited bioavailability and the difficulties in delivering these compounds across the blood-brain barrier is mentioned but warrants further exploration, particularly regarding the practicality and feasibility of using these compounds in human populations.
Furthermore, the manuscript predominantly focuses on preclinical studies, and there is a lack of emphasis on the clinical evidence supporting the effectiveness of kaempferol and quercetin in human AD patients. Including more data from clinical trials would strengthen the practical relevance of the proposed treatments and provide a clearer understanding of the potential side effects or limitations associated with their use.
In summary, while the manuscript provides a comprehensive and insightful review of the potential therapeutic benefits of kaempferol and quercetin in the context of AD, it would benefit from further exploration of the practical challenges in translating these findings into clinical applications. Additionally, a more robust discussion of the limitations and potential risks associated with the use of these phytochemicals in clinical settings is necessary for a more comprehensive understanding of their therapeutic potential.
Author Response
Reviewer 2 Comments:
The manuscript titled "Polyherbal and Multimodal: Kaempferol- and Quercetin-Rich Herbs Alleviate Symptoms of Alzheimer’s Disease" provides a comprehensive overview of the potential therapeutic effects of kaempferol and quercetin in the treatment of Alzheimer's Disease (AD). The paper follows the PRISMA guidelines for a systematic review and highlights the importance of a multimodal approach to address the complex pathogenesis of AD. The manuscript critically analyzes various aspects of the pathophysiology of AD, including the role of Aβ plaques, tau hyperphosphorylation, oxidative stress, neuroinflammation, and impaired synaptic plasticity. It effectively demonstrates how kaempferol and quercetin, two phytochemicals commonly found in various herbs, exhibit promising anti-AD effects through their antioxidant, anti-inflammatory, and neuroprotective properties. The strengths of this manuscript lie in its comprehensive approach to examining the mechanisms of action of kaempferol and quercetin, emphasizing their potential to alleviate multiple aspects of AD progression. The inclusion of relevant data and studies enhances the credibility of the proposed mechanisms and therapeutic potential of these phytochemicals. However, several notable limitations need consideration. While the manuscript extensively discusses the potential benefits of kaempferol and quercetin, it lacks a thorough discussion of the challenges associated with the translation of these findings into effective clinical treatments. The issue of limited bioavailability and the difficulties in delivering these compounds across the blood-brain barrier is mentioned but warrants further exploration, particularly regarding the practicality and feasibility of using these compounds in human populations. Furthermore, the manuscript predominantly focuses on preclinical studies, and there is a lack of emphasis on the clinical evidence supporting the effectiveness of kaempferol and quercetin in human AD patients. Including more data from clinical trials would strengthen the practical relevance of the proposed treatments and provide a clearer understanding of the potential side effects or limitations associated with their use. In summary, while the manuscript provides a comprehensive and insightful review of the potential therapeutic benefits of kaempferol and quercetin in the context of AD, it would benefit from further exploration of the practical challenges in translating these findings into clinical applications. Additionally, a more robust discussion of the limitations and potential risks associated with the use of these phytochemicals in clinical settings is necessary for a more comprehensive understanding of their therapeutic potential.
Response:
We appreciate your comments, and we have thoroughly explored all of the data regarding these limitations that we could find within the allotted timeframe for revisions to this manuscript. We have searched Google Scholar and PubMed for clinical trials regarding kaempferol and quercetin in AD patients, but we have not been able to find many beyond the trials already cited. Of note, we included a few more recent clinical trials in the discussion section, and their findings supported the initial clinical trial results discussed in the previous version of the manuscript. We have addressed this in the manuscript, and have explained potential reasons for this lack of clinical trials in general. Briefly, the clinical efficacy of kaempferol and quercetin in Alzheimer’s disease and related conditions could be due to their poor bioavailability to the brain. Thus, chemical modifications, such as nanoparticle loading, should be explored to overcome these challenges. These limitations were thoroughly explored in section 7 according to your concern. In short, we have determined that, despite the abundance of promising preclinical findings, the lack of clinical trials is likely due to the limited bioavailability to the brain, and chemical modifications or nanoparticle loading are necessary to send therapeutically relevant concentrations of the phytochemicals to the brain. We struggled to find limitations beyond what we have described in section 7.2, suggesting that kaempferol and quercetin are mostly safe for consumption.

Reviewer 3 Report
Comments and Suggestions for Authors
Dear Authors,
You attempted to summarize anti-AD properties of polyphenols Kaempferol and quercetin. Although intended work indicates a very important project, there are so many limitations to this work.
1. There are minor grammatical mistakes throught the manuscript. Please find them and consequently rectify them.
2. Authors have given detailed discussion of AD and its causes after introduction part. It is not necessary at all. It should written in a very concise way.
3. Mechanism of AD pathogenesis has been explained in a very detailed manner which must be concised to a small paragraph with necessary part.
4. Anti-AD activity of Kaempferol and Quercetin were explained very briefly without even mentioning the inhibition values. It is discussed in a vague manner. This part of the manuscript must be revamped completely and should be explained in detail.
5. In addition authors should also include comparative discussion of anti-AD properties of herbs (Kaempferol and Quercetin) and heterocyclic small molecules.
Comments on the Quality of English LanguageThere are minor grammatical mistakes in manuscript.
Author Response
Reviewer 3 Comments:
You attempted to summarize anti-AD properties of polyphenols Kaempferol and quercetin. Although intended work indicates a very important project, there are so many limitations to this work.
- There are minor grammatical mistakes throught the manuscript. Please find them and consequently rectify them.
Response:
We appreciate your observations and have accordingly corrected these mistakes. This may have changed some of the writing, but we believe that these changes ultimately alleviate the grammatical mistakes. You may notice that some of the paragraphs have been moderately rearranged and that there were changes to the wording in some places. Thus, sections 3-5 and section 7 are mostly color-coded in red to make all manuscript changes easily visible. We took this approach because we realized that, not only did our original setup of some paragraphs result in grammatical mistakes, but also, the previous arrangement of some paragraphs could be seen as random and led to repetitive wording. This also allowed us to nicely incorporate the newer references that we found during this revision process amongst the older references that we previously had. Notably, a few old references were removed from the manuscript. Subsections regarding insulin resistance were also removed to preserve the clarity of our proposed mechanisms. While this topic is still very important to the development of Alzheimer’s Disease treatments, we felt that it was ultimately beyond the scope of this review and had mostly become a distraction. Still, we feel that these rearrangements did not change the meaning of the manuscript, but rather, improved the clarity of it. In short, we felt that some paragraphs could be better organized to enhance the reader’s ability to follow the main points of the paper in sequential order. We feel that the new version of the manuscript perfectly reflects our proposed mechanisms in an organized fashion, and we thank you for your advice!
- Authors have given detailed discussion of AD and its causes after introduction part. It is not necessary at all. It should written in a very concise way. Mechanism of AD pathogenesis has been explained in a very detailed manner which must be concised to a small paragraph with necessary part.
Response:
We appreciate your comment, and we took your advice to consolidate the section about AD pathogenesis into a smaller section. While there was too much important information to consolidate into one short paragraph, we have cut the word count of the original section 3 by almost half. We believe that this greatly improved the manuscript and still emphasized the important anti-AD effects of kaempferol and quercetin.
- Anti-AD activity of Kaempferol and Quercetin were explained very briefly without even mentioning the inhibition values. It is discussed in a vague manner. This part of the manuscript must be revamped completely and should be explained in detail.
Response:
We have taken your advice and have sought out affinity values and applied it to a table featured in the text. We took this approach to enhance the ease of information gathering for the readers. The estimated affinity or potency of kaempferol/quercetin for each target were deduced from molecular docking studies (affinity) or from studies that elucidated the inhibitory constant (potency), and we chose to express the predicted affinities/potencies of the two flavonoids for their targets either as docking scores, binding energies (-kcal/mol) or IC50 values (uM), depending on how these were expressed in the cited literature. Binding energy (BE) measures the energy released from the molecular interaction between the ligand and target. Thus, the lower the binding energy, the higher the affinity (Pantsar and Poso, 2018). We have incorporated some of this data into a table (Table 1), which can be found in section 5.2 of the manuscript (of note, some proteins that were not specifically mentioned in section 5, but were mentioned elsewhere in the manuscript, were also included). In this table, we only cited the studies that were relevant to AD or related disorders/risk factors (such as major depressive disorder), and expressed the affinities as binding energies. Molecular docking scores (DS) are computational estimates of binding affinity based on the calculated molecular interaction based on the crystal structures of both the drug and receptor of interest. Molecular docking scores take into account several factors such as the Gibbs free energy, van der Waals forces, and H-bonding (Pantsar and Poso, 2018). We have also explored the limitations of molecular docking studies in the discussion section. Finally, IC50 values were provided to demonstrate the potencies of kaempferol and quercetin for inhibiting their targets, if the IC50 values were available from recent literature. IC50 values are not direct measures of affinity. However, high-affinity drugs typically produce high-potency effects. We explored Google Scholar and PubMed for articles clearly outlining the affinity of the phytochemicals for GSK-3β, AKT, PI3K, and other proteins relevant to this review, but have struggled to find an abundance of data from radioligand competition assays alone. We did, however, find a few studies outlining the inhibitory constant (IC50) values for some other proteins. All of these parameters were explained in the caption for Table 1. We hope that this table will clarify the strength of these proposed interactions.
- In addition authors should also include comparative discussion of anti-AD properties of herbs (Kaempferol and Quercetin) and heterocyclic small molecules.
Response:
We have considered your comment and have added a few similar flavonoids with heterocyclic structures to the discussion section of the manuscript. These molecules, although not a comprehensive list, included morin, luteolin, and rutin. These flavonoids had very similar effects to quercetin and kaempferol in reversing AD-related damage, including activating PI3K/AKT signaling [Carmona et al., 2018; Gur et al., 2021; He et al., 2021], increasing BDNF signaling [Mohammadi et al., 2021], and reversing aberrant Aβ and tau function [Mohammadi et al., 2021; Soubh et al., 2021; Sawmiller et al., 2014]. Of note, another difference between kaempferol and morin was briefly mentioned in section 4.2 of the original manuscript version that we submitted in October. We kept this comparison in that section as it fits best with the original aim of that paragraph.

Reviewer 4 Report
Comments and Suggestions for Authors
Review article on Polyherbal and Multimodal: Kaempferol- and Quercetin-Rich Herbs Alleviate Symptoms of Alzheimer’s Disease recommended for acceptance after minor revision.
Comment 1: Abstract and conclusion part is not correlated. Correct the sentences
Comment 2: is this Figure 1 necessary?
Comment 3: Include the structures of Kempferol and Quercetin. Mention their non specific inhibitory potential.
Comment 4: Cross check the molecular mechanism mentioned in
Figure 3 with recent sources.
Comment 5: Include recent references,
Comment 6: It has been known that both these compounds known to inhibit KAT and KDAC apart from Kinases. Brief about this inhibition and its connection with Aβ-mediated tau hyperphosphorylation.
Refer: Selvi RB, Swaminathan A, Chatterjee S, Shanmugam MK, Li F, Ramakrishnan GB, Siveen KS, Chinnathambi A, Zayed ME, Alharbi SA, Basha J, Bhat A, Vasudevan M, Dharmarajan A, Sethi G, Kundu TK. Inhibition of p300 lysine acetyltransferase activity by luteolin reduces tumor growth in head and neck squamous cell carcinoma (HNSCC) xenograft mouse model. Oncotarget. 2015 Dec 22;6(41):43806-18. doi: 10.18632/oncotarget.6245.
Comments on the Quality of English Language
Moderate editing of English language required
Author Response
Reviewer 4 Comments:
Review article on Polyherbal and Multimodal: Kaempferol- and Quercetin-Rich Herbs Alleviate Symptoms of Alzheimer’s Disease recommended for acceptance after minor revision.
Comment 1: Abstract and conclusion part is not correlated. Correct the sentences
Response:
We have corrected the conclusion and abstract to ensure that they are both relevant to each other, and we appreciate this observation! We feel that a few small details in the abstract were missing in the conclusion, so we have corrected this discrepancy to clarify the main points of the review article.
Comment 2: is this Figure 1 necessary?
Response:
We appreciate your consideration and your input. However, we have decided to keep Figure 1, but instead of remaining in the abstract, we have moved it to the discussion section, according to the suggestion of Reviewer #1, to reiterate the key concepts of this manuscript.
Comment 3: Include the structures of Kempferol and Quercetin. Mention their non specific inhibitory potential.
Response:
We have added Table 1, including the molecular docking scores that assess the possible affinity of the phytochemicals for the described targets, if applicable. We have described our rationale for using these studies in our rebuttal to reviewer #3. We hope that this will also clarify your concern regarding the strength of their interactions. These values included binding energies, docking scores, and IC50 values (potency) for any protein that we could find this data for. Other important proteins were included in the table and, if affinity values were not available, the interaction was noted as “indirect” under the column for affinity/potency.
Comment 4: Cross check the molecular mechanism mentioned in Figure 3 with recent sources. Comment 5: Include recent references,
Response:
We have sought out as many recent references as we could within the allotted time, and we have accordingly added them to the manuscript. These new references complemented our manuscript very well and reassured the relevance of our mentioned mechanisms.
Comment 6: It has been known that both these compounds known to inhibit KAT and KDAC apart from Kinases. Brief about this inhibition and its connection with Aβ-mediated tau hyperphosphorylation.
Refer: Selvi RB, Swaminathan A, Chatterjee S, Shanmugam MK, Li F, Ramakrishnan GB, Siveen KS, Chinnathambi A, Zayed ME, Alharbi SA, Basha J, Bhat A, Vasudevan M, Dharmarajan A, Sethi G, Kundu TK. Inhibition of p300 lysine acetyltransferase activity by luteolin reduces tumor growth in head and neck squamous cell carcinoma (HNSCC) xenograft mouse model. Oncotarget. 2015 Dec 22;6(41):43806-18. doi: 10.18632/oncotarget.6245.
Response:
We loved your recommendation to investigate the protective epigenetic mechanisms of kaempferol and quercetin, and have accordingly added these data to the manuscript. We think that this opened an exciting avenue for future research and further explained the neuroprotective effects of both flavonoids. We appreciated your recommended article, and we read through it. However, the article suggested that only luteolin (and neither kaempferol nor quercetin) could inhibit KAT. Still, we found a few recent studies that support this mechanism for kaempferol and quercetin and have included them in the manuscript. While it was somewhat difficult to find an abundance of recent studies, we believe that the few articles we found will greatly improve the manuscript according to your recommendations! These publications robustly supported the roles of kaempferol and quercetin in modulating KATs and KDACs to reduce neuroinflammation. However, we did not find studies that supported a direct effect of this mechanism on Aβ and tau. Thus, the main outcome of kaempferol and quercetin’s interactions with KATs and KDACs is likely anti-inflammatory. This would indirectly oppose the signaling of Aβ and hyperphosphorylated tau by resolving the neuroinflammatory environment that facilitates the accumulation of amyloid plaques and neurofibrillary tangles. We have already extensively discussed this connection in the manuscript, and we hope that these additions will also satisfy your recommendations.

Round 2
Reviewer 3 Report
Comments and Suggestions for Authors
Authors have considered all the comments with atmost care. Probably, the manuscript is suitable for publication.